# Revisiting the Activation Function for Federated Image Classification

## Abstract

Federated learning (FL) has become one of the most popular distributed machine learning paradigms; these paradigms enable training on a large corpus of decentralized data that resides on devices. The recent evolution in FL research is mainly credited to the refinements in training procedures by developing the optimization methods. However, there has been little verification of other technical improvements, especially improvements to the activation functions (e.g., ReLU), that are widely used in the conventional centralized approach (i.e., standard data-centric optimization). In this work, we verify the effectiveness of activation functions in various federated settings. We empirically observe that off-the-shelf activation functions that are used in centralized settings exhibit a totally different performance trend than do federated settings. The experimental results demonstrate that HardTanh achieves the best accuracy when severe data heterogeneity or low participation rate is present. We provide a thorough analysis to investigate why the representation powers of activation functions are changed in a federated setting by measuring the similarities in terms of weight parameters and representations. Lastly, we deliver guidelines for selecting activation functions in both a *cross-silo* setting (i.e., a number of clients $\leq$ 20) and a *cross-device* setting (i.e., a number of clients $\geq$ 100). We believe that our work provides benchmark data and intriguing insights for designing models FL models. The code is available at `https://anonymous.4open.science/r/FL_ACT-160B/`.

## 1 Introduction

Federated learning (FL) has become a common and ubiquitous paradigm for collaborative machine learning techniques (Bonawitz et al., 2019; Caldas et al., 2018; Kairouz et al., 2019; Li et al., 2020; 2019; Shokri & Shmatikov, 2015; McMahan et al., 2017; Smith et al., 2017) because it maintains data privacy. Each client (e.g., mobile devices or the entire business) communicates with the central server by transferring their trained model but not the data; all local updates are aggregated into a global server-side model. Although a centralized method enhances generalization by employing a large amount of training data, the features of the FL methods appear to differ from those of a centralized method owing to data non-IIDness, client resource capability, and model communication (Kairouz et al., 2021; McMahan et al., 2017; Zhao et al., 2018).

Most FL studies focus on improving the performance of the global model by applying a new regularizer in the optimization algorithm. For instance, a proximal term is attached to optimize the local update to enhance the method's stability (Acar et al., 2021; Karimireddy et al., 2020; Li et al., 2021; 2020), and hence the local model does not diverge from the global model. Some studies (Hsu et al., 2019; Lin et al., 2020; Wang et al., 2020a;b; Yurochkin et al., 2019) improve the aggregation step of the local models by weight averaging for the server model. When additional public or synthetic datasets are allowed, the server proofreads the weights of the models by utilizing their data distribution for balancing (Zhao et al., 2018; Jeong et al., 2018; Goetz & Tewari, 2020; Hao et al., 2021). Recently, there has been an increasing demand for the personalization of models according to the client. Jiang et al. (2019) and Fallah et al. (2020) attempt to train personalized models for each client with a few rounds of fine-tuning rather than focusing on the performance of the server model. In consideration of system heterogeneity (i.e., clients having different computational and communication capabilities), Avdiukhin & Kasiviswanathan (2021) mitigate model communication

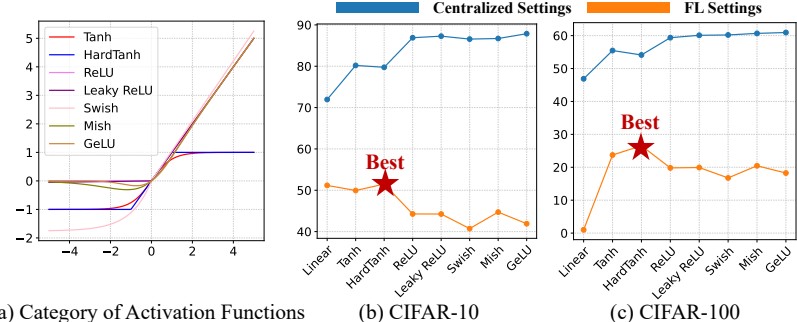

(a) Category of Activation Functions    (b) CIFAR-10    (c) CIFAR-100

Figure 1: (a) Plots of different activation functions. (b), (c) Accuracies on CIFAR-10 and CIFAR-100 according to the different activation functions, respectively. The blue line indicates the performance of models trained on a single central server. The orange line indicates the performance of models trained in an FL environment where 20 clients of the total 100 clients participate in the training per round. We use a model having four convolution layers and one classifier. Here, 'Linear' indicates a model without any activation function. In a single machine that centralizes the training data, the more up-to-date the activation function used, the better the performance (blue line). In contrast, interestingly, HardTanh (Collobert et al., 2011) prints the best server accuracy for both CIFAR-10 and CIFAR-100 under heterogeneous scenarios (orange line). A detailed explanation of the activation functions is provided in Appendix A.

problems by using asynchronous local stochastic gradient descent (SGD), and Horvath et al. (2021) improve accuracy for heterogeneous resource capacity by using different model sizes per client.

Despite the popularity of FL, some options for federated model optimization remain under-explored. Designing FL-familiar training recipes is essential to optimizing model performance, but few studies have attempted to design a new recipe instead of using those intended for a centralized setting. Charles et al. (2021) present an empirical analysis of the impact of hyperparameter settings for federated training dynamics from the perspective of a large cohort size. However, FL activation functions (McMahan et al., 2017; Karimireddy et al., 2020; Li et al., 2019) have rarely been studied, although activation functions play a crucial role in facilitating generalization and convergence. We thus raise the seemingly doubtful question: *Do activation functions that are popular in centralized settings also produce good optima in FL?*

To answer this question, we conduct a pilot experiment to compare performance in centralized settings with performance in FL settings. Figure 1 (b) and (c) show the accuracy of neural networks trained under a centralized setting and an FL setting according to the replacements in the activation function. Surprisingly, a neural network with Tanh has better accuracy than ReLU, which is a silver bullet in the field of centralized deep learning field. The problems mentioned above lead us to the intriguing question:

> *Do off-the-shelf activation functions that are intended for a centralized setting*
> *also perform appropriately in the FL setting?*

In this work, we answer the question with thorough empirical evaluations: *the most recently developed activation functions tend to degrade the performance of the server as the heterogeneity becomes more severe.*

Several considerations (e.g., the total number of clients, client participation, non-IIDness) in selecting activation functions may significantly improve the selection of activation function. Combining considerations may further boost the model accuracy. We experiment with various activation functions, including functions that are widely-used and rarely-used in the centralized setting, in various environments based on CIFAR-10 and CIFAR-100. The experiments identify an interesting phenomenon in FL, in which applying activation functions like ReLU in stacked convolutional layers demonstrates low accuracy owing to the shape of the function. We also provide an analysis of the representation power according to the different activation functions for federated image classification. Our key contributions are summarized as follows:

- We provide guidelines for selecting activation functions in FL. FL has the following special considerations: number of clients, participation ratio, and non-IIDness. We provide guide-

lines for *cross-silo* settings (i.e., for a number of clients $\leq 20$) and *cross-device* settings (i.e., for a number of clients $\geq 100$); the suitability of the activation function depends on the situation.

- We provide an explanation for the performance degradation (i.e., for the performance difference between centralized settings and FL settings) of activation functions that are preferred in a centralized setting. Specifically, we measure similarities in the weight parameters and representations, and visualize the landscape.

- We empirically show that the HardTanh activation function (Collobert et al., 2011) leads to a better optimum than other activation functions such as ReLU (Nair & Hinton, 2010), Leaky ReLU (Maas et al., 2013), and GeLU (Hendrycks & Gimpel, 2016) for a severe non-IID setting, a low participation rate, and using a large number of clients. Additionally, we provide benchmark data for activation functions in FL for various models.

## 2 RELATED WORK

### 2.1 ACTIVATION FUNCTIONS IN NEURAL NETWORKS

In deep learning, activation functions are inevitable for learning non-linear latent representations; an input signal is transformed into a non-linear output. Recent evolution has introduced an enhancement of representation power and lower computational costs. In the main experiments, the following non-linear activation functions are used: **Tanh** (LeCun et al., 2015a), **HardTanh** (Collobert et al., 2011), **ReLU** (Nair & Hinton, 2010), **Leaky ReLU** (Maas et al., 2013), **Swish** (Ramachandran et al., 2017), **Mish** (Misra, 2019), and **GeLU** (Hendrycks & Gimpel, 2016).

Activation functions have been devised for the gradient exploding/vanishing problem; where the magnitudes of gradients become either near zero or infinite during backward propagation. A general choice for activation functions has been ReLU, which enable efficient propagation. On the other side, ReLU is not differentiable at zero, and it causes a significant number of dying neurons by forgetting the information during propagation. Recent works (Ramachandran et al., 2017; Misra, 2019) have achieved smoother optima in centralized learning by designing self-regularized gradients; whereas those based on FL settings are badlands.

### 2.2 FL METHODS

Federated optimization methods manage to handle multiple clients without collecting data, and they use server weights from a central server to coordinate the global model across the network. In particular, these methods aim to minimize the following objective function:

$$\min_{w} f(w) \quad \text{where} \quad f(w) = \frac{1}{N} \sum_{k=1}^{N} f^{(k)}(w) \tag{1}$$

where $f^{(k)}$ is the loss function based on the client $k$. $N$ is the total number of clients. At each round, $K \ll N$ clients are selected from the total number of devices. The selected clients run each local model using SGD for $E$ number of local epochs, and they ultimately aggregate the selected models at the server model.

In the FL environment, the global model can be drifted by optimizing the local clients because the statistical data non-IIDness causes different local optima that are far apart from each other. This is called *client drift* (Karimireddy et al., 2020; Khaled et al., 2019; Reddi et al., 2020), and it indicates the inconsistency between optima. Recently, some works have prevented client drift by designing aggregation methods; Wang et al. (2020b) present a method of normalized averaging that removes objective inconsistency, and Zhang et al. (2021) propose a training algorithm for group knowledge transfer, which allows each client to keep a personalized prediction on the server to assist the local training of other clients. Federated Averaging (FedAvg (McMahan et al., 2017)) uses the local server for SGD for $E$ number of epochs. As a result, the selected client $k$'s weight is updated as $w_k$. To aggregate the local client models at each round, FedAvg sums and averages for the server model parameters formulated as:

$$w^t = \frac{1}{K} \sum_{k \in S_t} w_k^t \tag{2}$$

where $w^t$ is the server weight of the $t$-th round, $w_k^t$ is the client $k$'s weight after local training using $w^{t-1}$, and $S_t$ is the client set. McMahan et al. (2017) empirically show the significance of additionally tuning the hyperparameters in FL training. We present that with respect to the architectural and operational side. Additional details of related work are explained in Appendix A.

## 3 EXPERIMENTS

In this section, we compare several activation functions. We categorize the activation functions into two groups: (1) ReLU, Leaky ReLU, Swish, Mish, and GeLU as recent state-of-the-art (SOTA) activation functions that are widely used in centralized settings; and (2) Tanh and HardTanh as Tanh-like activation functions that are not widely used in centralized settings.

### 3.1 EXPERIMENTAL SETUP

**Dataset and non-IID Settings.** Two benchmark datasets are employed: CIFAR-10 and CIFAR-100 (Krizhevsky et al., 2009). We provide the descriptions of the datasets in Appendix B. To randomize the non-IID data, we assume that all client training data use class labels according to an independent categorical distribution of $N$ classes parameterized by the vector $q$:

$$q_i \geq 0, \ i \in [1, N] \quad \text{and} \quad \sum_{i \in [1, N]} q_i = 1$$

For the heterogeneous distribution, the *Dirichlet distribution* (Hsu et al., 2019; Yurochkin et al., 2019), $q \sim \text{Dir}(\boldsymbol{\alpha})$ is used, where $\boldsymbol{\alpha}$ is an $N$-length concentration vector having all elements $\alpha > 0$, that is, the prior distribution for $N$ classes controls the heterogeneity of clients.

**Models.** Our study focuses on compact models that are realistically possible in FL. Therefore, we mainly use a simple `ConvNet` having four convolutional layers and one classifier; `ConvNet4` refers to `ConvNet` with four convolutional layers. The first convolution layer has 64 kernels, and deeper layers have a larger number of kernels (O'Shea & Nash, 2015). For additional models, which have shortcut and batch normalization layers, we use `Resnet20`, `Resnet32`, `Resnet44` (He et al., 2016), and `MobileNetv2` (Sandler et al., 2018). The Details of settings and model architectures of `ConvNet` are provided in Appendix B.

**Training Details.** In this study, we conduct numerical experiments by changing the number of clients $N$, the client participation ratio $R$, and the Dirichlet distribution constant $\alpha$. We adapt FedAvg and perform 200 rounds with 5 local epochs using a learning rate of 0.01, with a learning decay of 0.1 at the 50th and 75th round, a weight decay of $1e-4$, and a momentum of 0.9. The number of clients available in different FL settings is limited; in a cross-silo setting, a small number of clients are available, and a large number of clients are requested in a cross-device. For the cross-silo setting, we use $N = 20$ and $R = 0.2$, which we select 4 clients at each round. For the cross-device setting, we use $N = 100$ and $R = 0.2$, which we select 20 clients at each round.

We mainly demonstrate the training of `ConvNet4` on CIFAR-10 heterogeneously distributed by modifying the $\alpha$ in the Dirichlet distribution and the client participation rate $R$. In the captions, we explain each $N$, $R$, and $\alpha$ value.

### 3.2 COMPARATIVE EXPERIMENTS ON THE CHANGES IN ACTIVATION FUNCTIONS

Table 1 shows the result of both centralized and FL settings using CIFAR-10 and CIFAR-100 as the datasets. With the centralized setting, GeLU shows the best performance, and other recent SOTA activation functions surpass the Tanh-like activation functions. However, with the FL setting, the activation functions show a significantly different tendency. HardTanh achieves the highest accuracy. Furthermore, the recent SOTA activation functions show lower accuracy than Linear using CIFAR-100 as the dataset. The activation functions show different accuracy drops; only the recent SOTA activation functions have an accuracy drop near 40, whereas HardTanh and Tanh have 26.21 and 28.42 on CIFAR-10. As a result, we can find that the most popular activation function, ReLU (as well as recent SOTA activation functions), does not show outstanding performance in an FL setting.

### 3.3 STRATEGIES FOR SELECTING ACTIVATION FUNCTIONS IN FL

This section presents the experimental results and guidelines for selecting the activation functions for various FL settings. FL settings have various environmental limitations relative to centralized settings. There have additional components to consider, such as the number of clients, non-IIDness, and the participation ratio.

Table 1: Server accuracy of `ConvNet4` in centralized and FL settings on two datasets (CIFAR-10, CIFAR-100). Centralized settings are used to train one server model using all training data, and FL settings are used to train 100 clients with non-IID data. We use $R = 0.2$, and $\alpha = 0.1$. For all tables afterwards we bold the highest accuracy except for Linear.

| Activation Function | CIFAR-10 | | CIFAR-100 | |
|---|---|---|---|---|
| | Centralized Setting | FL setting | Centralized Setting | FL Setting |
| Linear | $73.74\pm1.54$ | 10.00 | $46.52\pm0.31$ | 28.39 |
| Tanh | $81.00\pm0.89$ | 52.58 | $55.39\pm0.24$ | 30.75 |
| HardTanh | $80.64\pm0.77$ | **54.43** | $54.31\pm0.46$ | **31.76** |
| ReLU | $87.01\pm0.11$ | 48.37 | $59.75\pm0.33$ | 23.99 |
| Leaky ReLU | $87.30\pm0.03$ | 48.34 | $60.28\pm0.22$ | 24.04 |
| Swish | $86.50\pm0.08$ | 46.16 | $60.02\pm0.47$ | 21.55 |
| Mish | $86.38\pm0.29$ | 50.02 | $60.67\pm0.03$ | 24.98 |
| GeLU | $\mathbf{87.77\pm0.11}$ | 47.46 | $\mathbf{61.34\pm0.35}$ | 23.26 |

Table 2: Server accuracy of `ConvNet4` with five different number of clients $N$ (10, 20, 50, 100, 200). We use $\alpha = 0.1$ with $R = 0.2$.

| Activation Function | $N = 10$ | $N = 20$ | $N = 50$ | $N = 100$ | $N = 200$ |
|---|---|---|---|---|---|
| Tanh | 67.14 | 65.97 | **55.28** | 52.58 | 48.92 |
| HardTanh | 67.90 | **66.53** | 54.88 | **54.43** | **49.64** |
| ReLU | **69.26** | 63.39 | 52.88 | 48.37 | 41.57 |
| Leaky ReLU | **69.26** | 63.67 | 53.12 | 48.34 | 41.65 |

**Number of Clients.** For different FL strategies, the number of clients varies. A cross-silo setting uses fewer than 20 clients, and a cross-device setting use more than 100 clients. The total accuracy drop grows as the number of clients rises. Table 2 shows the accuracy with different client numbers. As the number of clients increases, the overall drop in accuracy increases. The Tanh-like activation functions surpass the recent SOTA activation functions with larger client numbers (20, 50, 100, 200). Additionally, the accuracy difference between the recent SOTA activation functions and Tanh-like activation functions gets larger as client numbers increase. Considering the observations for the number of clients, we hypothesize that as the number of clients increases, recent SOTA activation functions are increasingly affected and show a more significant accuracy drop.

**Non-IIDness.** With the Dirichlet distribution parameter $\alpha$, we can control the IID-ness of data: a larger value of $\alpha$ indicates lower non-IIDness (lower heterogeneity). Table 3 presents the accuracy for different values of $\alpha$. The overall reduction in accuracy rises as non-IIDness does. In most cases, HardTanh shows the highest accuracy. For 20 clients, the accuracy of the recent SOTA activation functions surpasses the Tanh-like activation functions at low non-IIDness. The shape of the recent SOTA activation functions with high non-IIDness causes a severe accuracy drop, which we discuss in section 4. For 100 clients, the Tanh-like activation functions are virtually unaffected by non-IIDness and outperforms the recent SOTA activation functions.

The low accuracy of the Tanh-like activation functions at $\alpha = 0.01$ occurs due to tough training settings and the Tanh-like activation functions fail to find an optimum, such as Linear. We conduct an additional experiment for $\alpha = 0.01$ with learning rate 0.005 to compare the Tanh-like activation functions and recent SOTA activation functions where the Tanh-like activation functions does not fail to train. Table 13 in Appendix C shows that Tanh-like activation functions surpass the recent SOTA activation functions without failure of finding optimum.

**Participation Ratio.** The participation of clients is limited in FL depending on the environment. In a cross-silo setting, high participation may be possible, whereas only limited participation is possible in a the cross-device setting. Table 4 shows the accuracy in FL settings for four different values of the participation ratio $R$. As the client participation decreases, the overall accuracy drop increases. For 100 clients, the Tanh-like activation functions achieve the highest accuracy. With 20 clients, however, there is a noticeably larger accuracy drop as participation decreases for the most recent SOTA activation functions. As a result, as the participation ratio decreases, the accuracy of the Tanh-like activation functions reverses recent SOTA activation functions. With increased

Table 3: Server accuracy of `ConvNet4` with four different Dirichlet constant values $\alpha$ (0.01, 0.1, 1, 10). We use $N = 100$ and $N = 20$ with $R = 0.2$.

| Activation Function | $N = 100$ | | | | $N = 20$ | | | |
|---|---|---|---|---|---|---|---|---|
| | $\alpha = 10$ | $\alpha = 1$ | $\alpha = 0.1$ | $\alpha = 0.01$ | $\alpha = 10$ | $\alpha = 1$ | $\alpha = 0.1$ | $\alpha = 0.01$ |
| Linear | 62.48 | 62.43 | 10.00 | 10.00 | 66.48 | 66.48 | 63.54 | 10.00 |
| Tanh | 64.49 | 64.14 | 52.58 | 29.50 | 70.22 | 69.55 | 65.97 | 27.64 |
| HardTanh | **65.27** | **65.40** | **54.43** | 30.09 | 70.53 | 70.01 | **66.53** | 28.92 |
| ReLU | 57.80 | 56.23 | 48.37 | 34.03 | 75.59 | 74.21 | 63.39 | 34.70 |
| Leaky ReLU | 57.85 | 56.16 | 48.34 | 33.92 | 75.36 | 74.23 | 63.67 | 35.15 |
| Swish | 52.62 | 51.48 | 46.16 | 35.65 | 72.58 | 71.58 | 64.57 | 36.93 |
| Mish | 57.30 | 55.08 | 50.02 | **38.94** | 73.69 | 72.95 | 66.47 | **37.89** |
| GeLU | 55.59 | 54.34 | 47.46 | 36.09 | **75.68** | **74.41** | 65.62 | 37.77 |

Table 4: Server accuracy of `ConvNet4` with four different participation ratios $R$ (0.1, 0.2, 0.3, 0.4). We use $N = 100$ and $N = 20$ with $\alpha = 0.1$.

| Activation Function | $N = 100$ | | | | $N = 20$ | | | |
|---|---|---|---|---|---|---|---|---|
| | $R = 0.4$ | $R = 0.3$ | $R = 0.2$ | $R = 0.1$ | $R = 0.4$ | $R = 0.3$ | $R = 0.2$ | $R = 0.1$ |
| Linear | 10.00 | 10.00 | 10.00 | 10.00 | 65.85 | 66.12 | 63.54 | 58.36 |
| Tanh | **62.61** | 61.79 | 52.58 | **46.61** | 69.22 | 67.75 | 65.97 | **62.67** |
| HardTanh | 59.45 | **61.75** | **54.43** | 41.90 | 69.25 | 68.80 | **66.53** | 43.96 |
| ReLU | 53.26 | 50.67 | 48.37 | 41.95 | **71.96** | 70.17 | 63.39 | 54.47 |
| Leaky ReLU | 53.17 | 50.76 | 48.34 | 42.05 | 66.12 | **70.22** | 63.67 | 54.46 |
| Swish | 51.72 | 49.45 | 46.16 | 40.00 | 70.36 | 67.99 | 64.57 | 53.74 |
| Mish | 56.67 | 52.80 | 50.02 | 43.37 | 71.30 | 69.34 | 66.47 | 60.38 |
| GeLU | 53.35 | 50.61 | 47.46 | 41.50 | 72.32 | 70.10 | 65.62 | 54.05 |

client participation, client drift diminishes and the impact of the current SOTA activation functions' accuracy reduction decreases.

Different FL settings components affect accuracy when different activation functions are used, according to the observations above. The number of clients is the most dominant component, and it interacts with the effect of other components when a small number of clients is used. Therefore, the Tanh-like activation functions are favorable for a large number of clients, such as in a cross-device setting. For a cross-silo setting, for a data with high data non-IIDness, and for a low ratio of client participation, the Tanh-like activation functions are preferred. Conversely, the recent SOTA activation functions are preferred for a cross-silo setting, for data with low data non-IIDness, and for a high ratio of client participation.

### 3.4 ADDITIONAL EXPERIMENT

We implement additional experiments with a different FL method and models. We choose FedProx as the additional FL method and `Resnet20`, `Resnet32`, `Resnet44`, and `MobileNetv2` as the additional models. For the additional models, we only use ReLU and Leaky ReLU from the set of recent SOTA activation functions. For more experimental results, refer to Appendix C.

**FedProx.** As mentioned in section 1, studies to improve the server model in FL add proximal terms to the local object to enhance the method's stability so that the local model rarely differs from the global model. We choose FedProx (Li et al., 2020) as an additional FL method because it is the most basic method that improves the server model by adding a proximal term. The details of FedProx are shown in Appendix A. Table 5 shows the accuracy for different activation functions using FedProx as a learning algorithm. Similar to FedAvg, HardTanh achieves the best accuracy, followed by the other Tanh-like activation functions. Proximal terms in local training do not appear to help prevent accuracy loss.

Table 5: Server accuracy of `ConvNet4` using FedProx with the following settings: $N = 100$, $R = 0.2$, and $\alpha = 0.1$.

| Act. Func. | Acc. |
|---|---|
| Linear | 48.74 |
| Tanh | 45.50 |
| HardTanh | **49.16** |
| ReLU | 42.19 |
| Leaky ReLU | 42.40 |
| Swish | 35.79 |
| Mish | 39.83 |
| GeLU | 36.00 |

Table 6: Server accuracy of `Resnet` using four different Dirichlet constants, $\alpha$ (10, 1, 0.1, 0.01), with a fixed client participation ratio, $R = 0.2$, and four different client participation ratios, $R$ (0.4, 0.3, 0.2, 0.1) with a fixed Dirichlet constant, $\alpha = 0.1$. We use $N = 100$ with CIFAR-10 as the dataset.

| | Activation Function | $\alpha$ | | | | $R$ | | | |
|---|---|---|---|---|---|---|---|---|---|
| | | $\alpha = 10$ | $\alpha = 1$ | $\alpha = 0.1$ | $\alpha = 0.01$ | $R = 0.4$ | $R = 0.3$ | $R = 0.2$ | $R = 0.1$ |
| Resnet20 | Linear | 56.93 | 56.96 | 45.73 | 26.70 | 52.09 | 49.35 | 45.73 | 38.60 |
| | Tanh | **59.66** | 57.42 | **46.58** | 26.50 | 52.92 | 49.69 | **46.58** | 38.83 |
| | HardTanh | 59.60 | **57.54** | 46.19 | **26.52** | **53.69** | **50.16** | 46.19 | **39.28** |
| | ReLU | 56.90 | 56.95 | 45.35 | 25.73 | 52.54 | 48.71 | 45.35 | 37.53 |
| | Leaky ReLU | 56.78 | 56.49 | 45.18 | 26.10 | 52.23 | 48.38 | 45.18 | 37.71 |
| Resnet32 | Linear | 60.12 | 56.84 | 43.01 | 24.98 | 49.49 | 46.15 | 43.01 | 37.39 |
| | Tanh | **60.03** | 56.81 | 43.92 | 25.90 | **51.03** | **47.77** | 43.92 | 38.24 |
| | HardTanh | 59.30 | **57.39** | **43.99** | 25.51 | 50.75 | 47.73 | **43.99** | **38.34** |
| | ReLU | 58.52 | 56.34 | 43.09 | **26.66** | 49.42 | 46.49 | 43.09 | 37.87 |
| | Leaky ReLU | 58.97 | 56.16 | 43.42 | 26.42 | 49.40 | 46.81 | 43.42 | 37.33 |
| Resnet44 | Linear | 57.55 | 55.98 | 45.14 | 27.50 | 51.25 | 47.78 | 45.14 | 39.56 |
| | Tanh | 57.94 | 55.54 | **46.42** | **27.23** | 52.89 | 50.67 | **46.42** | 38.19 |
| | HardTanh | **58.80** | **57.54** | 45.98 | 26.46 | **53.95** | **51.23** | 45.98 | **38.88** |
| | ReLU | 56.89 | 54.23 | 44.42 | 26.06 | 50.31 | 46.25 | 44.42 | 38.57 |
| | Leaky ReLU | 57.02 | 53.76 | 43.97 | 26.37 | 50.40 | 46.84 | 43.97 | 37.35 |

**Resnet.** Table 6 shows the result of `Resnet20`, `Resnet32`, and `Resnet44` with four different values of $\alpha$ and $R$. The Tanh-like activation functions acheive better accuracy than the recent SOTA activation functions do. Even when employing models with batch normalization and shortcut layers, the Tanh-like activation functions outperform the recent SOTA activation functions. For $\alpha = 0.01$ and using `Resnet32` as the model, the Tanh-like activation functions fails to find the optimum and perform poorly.

**MobileNetv2.** Table 7 shows the result of `MobileNetv2` with four different values of $R$ for $N = 100$ and $\alpha = 0.1$. Tanh-like activation functions surpass recent SOTA activation functions. In particular, the accuracy of the recent SOTA activation functions is lower than Linear. As the participation ratio rises, the recent SOTA activation functions surpass the Tanh-like activation functions on the CIFAR-10 dataset, demonstrating the same result as Table 4.

Table 7: Server accuracy of `MobileNetv2` with four different participation ratios, $R$ (0.1, 0.2, 0.3, 0.4). We use $N = 100$ with $\alpha = 0.1$.

| Activation Function | CIFAR-10 | | | | CIFAR-100 | | | |
|---|---|---|---|---|---|---|---|---|
| | $R = 0.4$ | $R = 0.3$ | $R = 0.2$ | $R = 0.1$ | $R = 0.4$ | $R = 0.3$ | $R = 0.2$ | $R = 0.1$ |
| Linear | 36.74 | 36.86 | 35.86 | 35.50 | 15.02 | 14.76 | 14.23 | 13.17 |
| Tanh | 33.39 | 33.64 | **31.59** | **31.83** | **19.20** | **16.99** | **14.76** | **11.19** |
| HardTanh | 33.82 | 32.41 | 28.91 | 29.95 | 18.64 | 15.91 | 13.54 | 9.39 |
| ReLU | **36.99** | **34.21** | 26.93 | 20.12 | 16.71 | 13.97 | 8.16 | 5.17 |
| Leaky ReLU | 35.54 | 33.07 | 30.69 | 20.44 | 16.92 | 13.84 | 10.14 | 5.67 |

# 4 ANALYSIS

We present thorough investigations of model behavior and the changes in representation during the local training to answer the question: *Do recent SOTA activation functions have a disadvantage in an FL setting?*

## 4.1 INVESTIGATION OF WEIGHT PARAMETERS AND LATENT REPRESENTATIONS

The shape of the activation function varies accuracy in FL setting. The activation function selects important features to pass through each layer, and the model is trained using these features. According to Figure 2 (a), the number of selected features varies according to the shape of the activation function. In a conventional centralized setting, a single model can access all the data and select optimal training features. However, in an FL setting, each client can only access a portion of the data, which is partitioned in the non-IID condition, and each client trains its model to select features that are important to itself. This results in a phenomenon known as client drift.

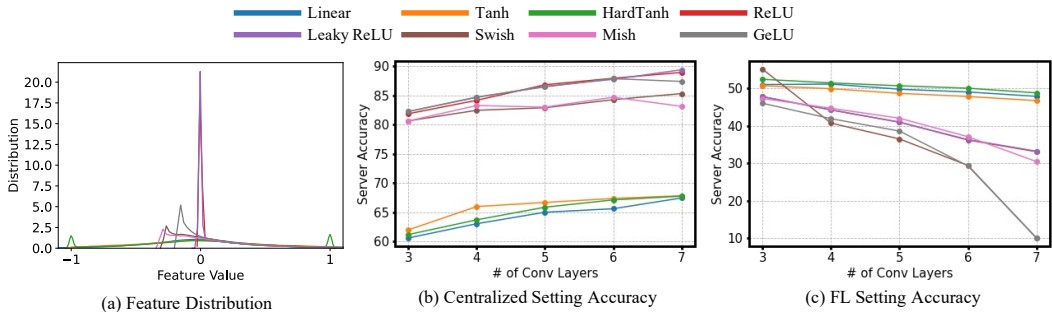

Figure 2: (a) demonstrates the feature distribution of all class 0 images in the CIFAR-10 test dataset after passing through the first convolution layer and its activation function. After passing the convolution layer and activation function, we flatten the feature values and draw a distribution. (b) shows the accuracy of `ConvNet` in centralized setting with different widths and depths. (c) shows the accuracy of `ConvNet` in federated setting with different widths and depths. For all three figures, we use $N = 100$, $R = 0.2$ and $\alpha = 0.1$.

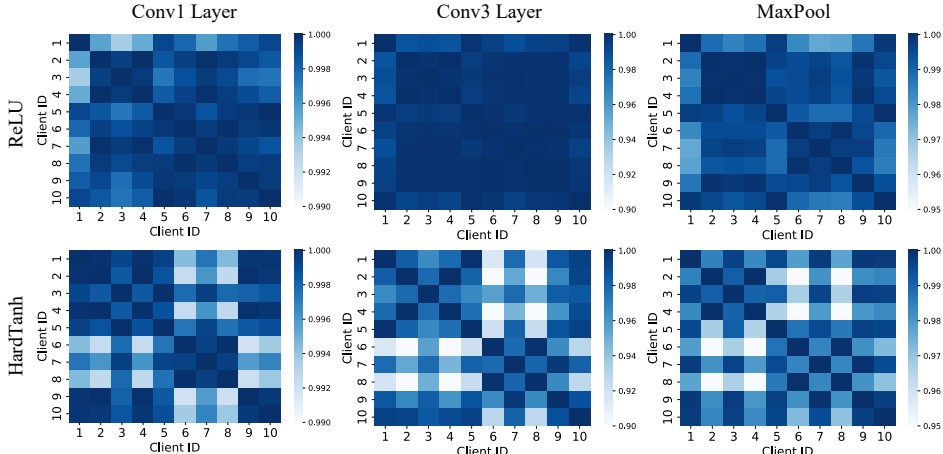

Figure 3: CKA similarity between 10 client using test images of CIFAR-10. Each client's model is the model before $100th$ aggregation. We use $N = 100$ with $R = 1.0$ and $\alpha = 0.1$ for training. We calculate the CKA similarity using features passing through each layer and its activation function.

During the FL aggregation step, a problem arises where important features for the global optimum cannot be selected due to client drift. This phenomenon appears to be severe when the recent SOTA activation functions are used. Due to the shape of their activation functions, the excluded features are greater in number than for the Tanh-like activation functions, and a severe accuracy drop occurs. In addition, when heterogeneity increases, selection failure of features using the recent SOTA activation functions achieves its peak, and this phenomenon reaches its pinnacle. This can be summarized simply by saying that the Tanh-like activation functions have low sensitivity to the accuracy drop in the FL aggregation step because they exclude a much smaller number of features than do the recent SOTA activation functions. In this aspect, deeper models that achieve greater accuracy in a centralized context perform poorly in a FL situation because they reject a greater number of features as the model gets deeper. In Figure 2 (c), deeper ConvNets with recent SOTA activation functions in a FL setting demonstrate a significant reduction in accuracy, whereas Figure 2 (b) demonstrates an accuracy improvement for deeper Convnets in a centralized setting.

This perspective can be used to explain the empirical results in section 3. A greater number of clients, a high data non-IIDness, and a low client participation rate all contribute to a high level of heterogeneity. In Table 2, as the number of clients increases, in Table 3 with N = 20, as the dirichlet constant decreases, and in Table 4, as the participation rate R decreases, heterogeneity increases, and as a result, the recent SOTA activation function demonstrates a significant drop in accuracy and is surpassed by Tanh-like activation functions.

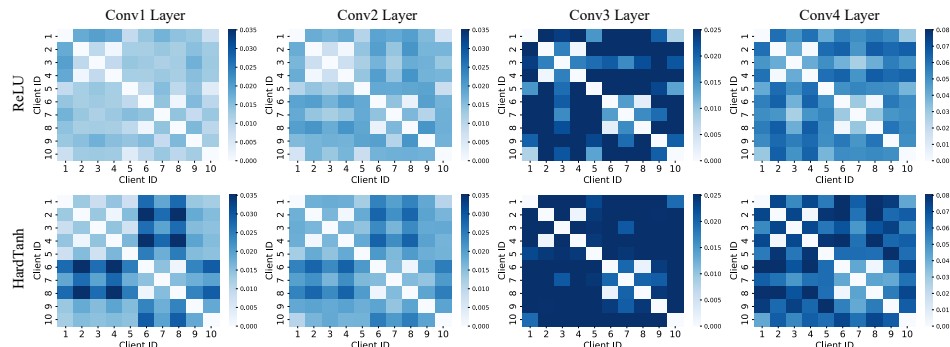

Figure 4: Weight difference between 10 client selected in Figure 3. We calculate the weight difference by subtracting each client's weight from the same layer and normalizing it with $L_2$norm.

To observe how the activation function affects `ConvNet4`, we perform two additional experimental studies on heterogeneous local models. For simplicity, we look at 10 clients out of a total of 100 clients. First, we perform centered kernel alignment (CKA) (Kornblith et al., 2019) to measure the similarity of the output features between different clients. Each client trains the server model for 5 epochs with their own non-IID data. In Figure 3, ReLU has a higher CKA similarity than does HardTanh in every layer. Because feature selection fails for the server model, as indicated above, ConvNet4 with ReLU has a smaller feature change than does HardTanh, which indicates a lower learning ability. Second, we calculate the weight differential to check how similar the clients weights are to each other. Figure 4 reveals that ConvNet4 with ReLU has a smaller weight difference in each layer than does HardTanh, and also indicates that ReLU has a limited learning ability.

## 4.2    ANALYSIS OF LANDSCAPE

We visualize the 2-D landscape of Tanh, HardTanh, ReLU, and Leaky ReLU. Figure 5 shows the 2-D landscape of each activation function. The color bar beside each figure shows the loss difference. Tanh and HardTanh have a number that is $10,000$ times smaller than ReLU and Leaky ReLU. This emphasizes that Tanh and HardTanh have a considerably smoother landscape than ReLU and Leaky ReLU.

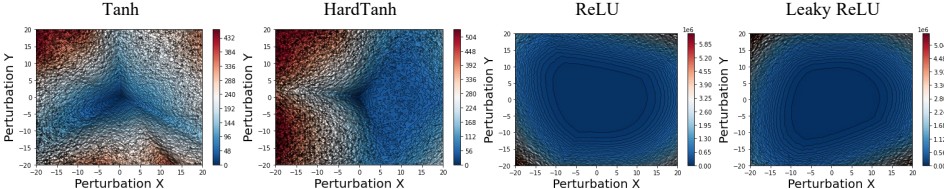

Figure 5: Landscape of `ConvNet4` with Tanh, HardTanh, ReLU and Leaky ReLU. We use $N = 100$, $R = 0.1$ and $\alpha = 0.01$ for training. We draw landscape with 150 levels.

These findings suggest that the sensitivity of the activation functions in the FL aggregation step, where the accuracy drop occurs, is indicated by the shape of the activation functions. The accuracy drop reaches its peak as the number of features excluded by an activation function increases, and it reaches its maximum with severe client drift: a large number of clients, a low client participation ratio, and high data non-IIDness.

## 5    CONCLUSION

This study clarifies that the drop in accuracy varies according to the activation function in FL. Our key finding is that the accuracy of the recent SOTA activation functions drops in an FL setting due to the shape of the functions, and HardTanh outperforms other activation functions in most environments. Additionally, we provide guidelines and benchmark data for selecting activation functions in various FL settings.

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

# Appendix

## A  RELATED WORK

### A.1  ACTIVATION FUNCTIONS IN NEURAL NETWORKS

In deep neural networks, using activation functions is a ubiquitous technique for learning non-linear latent representations; an input signal is transformed into the non-linear output centered on zero. Recent evolution occurs along with the enhancement of representation power and the efficiency of computational costs. In the experiment, we use the following non-linear activation functions:

**Tanh.**  A hyperbolic tangent function that is a smooth zero-centered function. The equation of Tanh is:

$$\text{Tanh}(x) = \frac{e^x - e^{-x}}{e^x + e^{-x}}$$

It is known that it is zero-centered, but computationally expensive and causes vanishing gradient problems as neural networks become deeper.

**HardTanh.**  Another variant of Tanh that involves lower computational costs owing to its piece-wise linearity. It is another variant of the hyperbolic tangent function, which represents computationally more efficient form of tanh:

$$\text{HTanh}(x) = \min(1, \max(-1, x))$$

**ReLU.**  An element-wise threshold operation on each input element where values less than zero are set to zero or otherwise used as is. The equation of ReLU (Hahnloser et al., 2000; Jarrett et al., 2009; Nair & Hinton, 2010) is:

$$\text{ReLU}(x) = \max(0, x)$$

ReLU is the abbreviation of *Rectified Linear Unit*, a modified linear function. When the input value of ReLU is negative, the gradient of its output value is zero; the model does not learn. ReLU has shown great performance with Convolutional Neural Networks (CNN) (Krizhevsky et al., 2012; LeCun et al., 2015b). Since ReLU is computationally cheap, it is still commonly used regardless of numerous attempts to replace it (Maas et al., 2013; Goodfellow et al., 2013; He et al., 2015; Clevert et al., 2015; Klambauer et al., 2017; Elfwing et al., 2018).

**Leaky ReLU.**  A variant of ReLU that is designed to prevent the *Dying Neuron* problem, where gradients will not be zero at any time during training. The equation of Leaky ReLU (Maas et al., 2013) is:

$$\text{LReLU}(x) = \max(0.01x, x)$$

It is ReLU multiplied by a tiny constant on the negative part. Due to the small range, the graphs are drawn almost similarly with ReLU.

**Swish.**  An unbounded function that uses the sigmoid function in its formula. The curve is similar to ReLU, but has is smooth and non-monotonic. The equation of Swish (Ramachandran et al., 2017) is:

$$\text{Swish}(x) = x \cdot sigmoid(x)$$

This function shows better accuracy than ReLU in deep neural networks regardless of batch size.

**Mish.**  A self-regularized non-monotonic activation function. The curve is similar to Swish, but it has a stronger regularization effect and a smoother gradient. The equation of Mish (Misra, 2019) is:

$$\text{Mish}(x) = x \cdot tanh(ln(1 + e^x)))$$

This function has the characteristic of allowing gradients to flow better than the Relu Zero Bound because it allows some negative numbers.

**GeLU.**  A Gaussian distribution-involved smooth variation on ReLU. It is often used in NLP tasks and vision tasks with Vision Transformer (Dosovitskiy et al., 2020) models. The equation of GeLU (Hendrycks & Gimpel, 2016) is:

$$\text{GeLU}(x) = x \cdot \frac{1}{2}[1 + tanh[\sqrt{2\pi}(x + 0.044715x^3)]]$$

GeLU is derived by combining the characteristics of dropout, zoneout, and ReLU.

## A.2 Algorithms of Federated Learning Methods

We use both FedAvg and FedProx for federated learning methods. Algorithm 1 shows the algorithm of FedAvg and Algorithm 2 shows the algorithm of FedProx. FedProx is similar to FedAvg in that it selects a selection of clients at each round, performs local training, and then averages client's weight to generate a global update. However, the difference between FedAvg and FedProx is shown in line 6. For local training, FedAvg trains each client's model using SGD with its local data whereas FedProx, trains each client with additional proximal term, $\frac{\mu}{2} \|w - w^t\|^2$. Using the proximal term which contributes to the method's stability by efficiently reducing the impact of variable modifications.

---

**Algorithm 1** Federated Averaging (FedAvg)

---

1: **Input:** $K, T, \eta, E, w^0, N, k \in [1, \cdots, N]$
2: **for** $t = 0, \cdots, T-1$ **do**
3:     Server selects a subset $S_t$ randomly which includes number of $K$ devices
4:     Server send $w^t$ to all selected devices
5:     **for** $i = 0, \cdots, E-1$ **do**
6:         Selected device $k \in S_t$ updates their local weight $w_k^{t+1}$ using SGD with step-size $\eta$
7:     **end for**
8:     Selected device $k \in S_t$ sends their local weight $w_k^{t+1}$ back to the server
9:     Server aggregates the local weights, $w_k^{t+1}$, and gets new server weight $w^{t+1} = \frac{1}{K} \sum_{k \in S_t} w_k^t$
10: **end for**

---

---

**Algorithm 2** FedProx

---

1: **Input:** $K, T, \eta, \mu, E, w^0, N, k \in [1, \cdots, N]$
2: **for** $t = 0, \cdots, T-1$ **do**
3:     Server selects a subset $S_t$ randomly which includes number of $K$ devices
4:     Server send $w^t$ to all selected devices
5:     **for** $i = 0, \cdots, E-1$ **do**
6:         Selected device $k \in S_t$ updates their local weight
        $w_k^{t+1} \approx \min_{w} h^{(k)}(w; w^t) = f^{(k)}(w) + \frac{\mu}{2} \|w - w^t\|^2$ with step-size $\eta$
7:     **end for**
8:     Selected device $k \in S_t$ sends their local weight $w_k^{t+1}$ back to the server
9:     Server aggregates the local weights, $w_k^{t+1}$, and gets new server weight $w^{t+1} = \frac{1}{K} \sum_{k \in S_t} w_k^t$
10: **end for**

---

# B Implementation Details

## B.1 Model Architecture

Figure 6 shows `ConvNet` with five different depth (3,4,5,6, and 7). Each version of `ConvNet` has convolution layer corresponding to the number after the model (e.g., `ConvNet3` has three convolution layers and `ConvNet7` has seven convolution layers). The details of each convolution layer is shown in Table 8. `ConvNet` with varying depth employ convolution layers in the order shown in Table 8 (e.g., `ConvNet3` use Conv1, Conv2, and Conv3 whereas, `ConvNet7` use Conv1, Conv2, Conv3, Conv4, Conv5, Conv6, and Conv7).

## B.2 Dataset statistics

We use both CIFAR-10 and CIFAR-100 for our experiments. As shown in Table 9, CIFAR-10 consists of 60000 images of size 32×32. It is divided into 10 classes, and each class consists of 6000 images. Also, each class has 5,000 training images and 1,000 test images. CIFAR-100 also consists of 60000 images with a size of 32×43. It is classified into 100 classes, and each class consists of 600 images. These 100 classes are divided into 20 superclasses, which we do not use in our experiments. Also, each class has 500 training images and 100 test images.

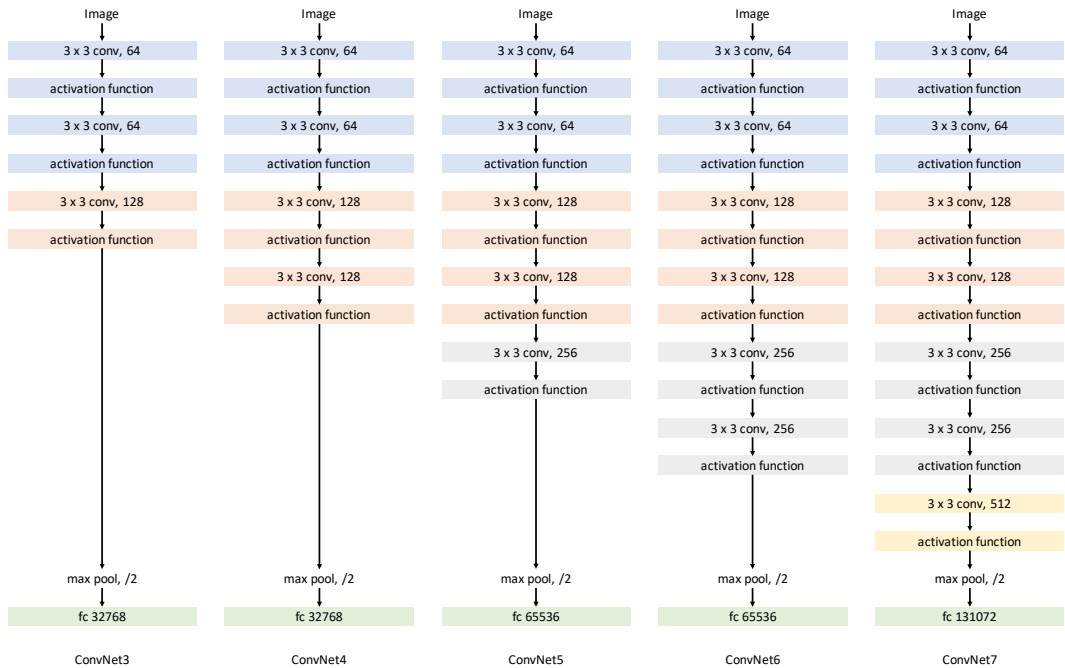

Figure 6: The architecture of `ConvNet` with different number of convolution layers.

Table 8: Setting of each convolution layer in `ConvNet`. Each version of `ConvNet` uses the number convolution layer as much as their version number. (i.e `ConvNet3` uses conv1 through conv3 and `ConvNet7` uses conv1 through conv7.)

| Layer | Number of Input Filter | Number of Output Filter | Kernel Size | Padding |
|-------|------------------------|-------------------------|-------------|---------|
| Conv1 | 3   | 64  | 3×3 | 1 |
| Conv2 | 64  | 64  | 3×3 | 1 |
| Conv3 | 64  | 128 | 3×3 | 1 |
| Conv4 | 128 | 128 | 3×3 | 1 |
| Conv5 | 128 | 256 | 3×3 | 1 |
| Conv6 | 256 | 256 | 3×3 | 1 |
| Conv7 | 256 | 512 | 3×3 | 1 |

Table 9: Dataset statistics.

| Data set | Train examples | Test examples | Class Number | Task |
|----------|----------------|---------------|--------------|------|
| CIFAR-10  | 50,000 | 10,000 | 10  | Image Classification |
| CIFAR-100 | 50,000 | 10,000 | 100 | Image Classification |

## C  ADDITIONAL EXPERIMENT RESULT

### C.1  CONVNET4 RESULT

Experiment results in section 3 fix variables such as $N$, $R$, and $\alpha$. Table 10 shows the result with all combinations of $R$ and $\alpha$ with $N = 20$. Table 11 shows the result with all combinations of $R$ and $\alpha$ with $N = 100$. Table 12 shows the result using CIFAR-100 as the dataset with all combinations of $R$ and $\alpha$ with $N = 100$. Comparing Table 10 and Table 11, we can see that with bigger client number, the Tanh-like activation function surpasses the recent SOTA activation functions. Additionally, with more complex dataset (CIFAR-100) shown in Table 12, the Tanh-like activation functions outperform

the recent SOTA activation functions with bigger accuracy gap. The low accuracy of the Tanh-like activation functions using CIFAR-10 as dataset with $\alpha = 0.01$ occurs due to severe tough training settings and fails to find the optimum.

To checkout the performance when the Tanh-like activation functions does not fail to find the optimum, we use an additional setting with learning rate 0.005. Table 13 shows the accuracy of `ConvNet4` with fixed dirichlet constant $\alpha = 0.01$, which the Tanh-like activation functions failed to find optimum in Table 10 and Table 11. The Tanh-like activation functions surpass the recent SOTA activation functions in all conditions. In addition, the Tanh-like activation functions surpass the highest accuracy with dirichlet constant $\alpha = 0.01$ of the recent SOTA activation functions presented in Table 10 and Table 11.

Table 10: Server accuracy of `ConvNet4` using four different $\alpha$ and four different participation $R$, where $N = 20$. For different dirichlet distribution constant, $\alpha = 0.01$ is the most non-IID setting and $\alpha = 10$ is the most IID setting. Since using Linear at $\alpha = 0.01$ with participation ratio 0.1, 0.2, 0.3 could not train, leave blank at $\alpha = 0.01 \rightarrow 0.1$ with participation ratio 0.1, 0.2, 0.3. The most right columns show the accuracy drop as non-IIDness increases.

| Participation Ratio | Activation Function | $\alpha = 10$ | $\alpha = 1$ | $\alpha = 0.1$ | $\alpha = 0.01$ | $\alpha = 10 \rightarrow 1$ | $\alpha = 1 \rightarrow 0.1$ | $\alpha = 0.1 \rightarrow 0.01$ |
|---|---|---|---|---|---|---|---|---|
| 0.1 | Linear | 64.76 | 64.48 | 58.36 | 10.00 | 0.28 | 6.12 | - |
| | Tanh | 66.83 | 66.40 | **62.67** | 22.44 | 0.43 | 3.73 | 40.23 |
| | HardTanh | 67.42 | 66.90 | 43.96 | 22.61 | 0.52 | 22.94 | 21.35 |
| | ReLU | 69.27 | 67.40 | 54.47 | 27.22 | 1.87 | 12.93 | 27.25 |
| | Leaky ReLU | **69.29** | 67.39 | 54.46 | 28.49 | 1.90 | 12.93 | 25.97 |
| | Swish | 66.33 | 64.56 | 53.74 | 30.48 | 1.77 | 10.82 | 23.26 |
| | Mish | 69.00 | **67.58** | 60.38 | **33.13** | 1.42 | 7.20 | 27.25 |
| | GeLU | 68.52 | 66.36 | 54.05 | 30.71 | 2.16 | 12.31 | 23.34 |
| 0.2 | Linear | 66.48 | 66.48 | 63.54 | 10.00 | 0.00 | 2.94 | - |
| | Tanh | 70.22 | 69.55 | 65.97 | 27.64 | 0.67 | 3.58 | 38.33 |
| | HardTanh | 70.53 | 70.01 | **66.53** | 28.92 | 0.52 | 3.48 | 37.61 |
| | ReLU | 75.59 | 74.21 | 63.39 | 34.70 | 1.38 | 10.82 | 28.69 |
| | Leaky ReLU | 75.36 | 74.23 | 63.67 | 35.15 | 1.13 | 10.56 | 28.52 |
| | Swish | 72.58 | 71.58 | 64.57 | 36.93 | 1.00 | 7.01 | 27.64 |
| | Mish | 73.69 | 72.95 | 66.47 | **37.89** | 0.74 | 6.48 | 28.58 |
| | GeLU | **75.68** | **74.41** | 65.62 | 37.77 | 1.27 | 8.79 | 27.85 |
| 0.3 | Linear | 61.15 | 67.54 | 66.12 | 10.00 | -6.39 | 1.42 | - |
| | Tanh | 71.95 | 71.83 | 67.75 | 30.01 | 0.12 | 4.08 | 37.74 |
| | HardTanh | 72.33 | 71.49 | 68.80 | 30.51 | 0.84 | 2.69 | 38.29 |
| | ReLU | 78.65 | 77.77 | 70.17 | 41.00 | 0.88 | 7.60 | 29.17 |
| | Leaky ReLU | 78.62 | 78.13 | **70.22** | 41.31 | 0.49 | 7.91 | 28.91 |
| | Swish | 76.25 | 75.54 | 67.99 | 43.19 | 0.71 | 7.55 | 24.80 |
| | Mish | 77.32 | 76.46 | 69.34 | **45.16** | 0.86 | 7.12 | 24.18 |
| | GeLU | **79.19** | **78.18** | 70.10 | 44.98 | 1.01 | 8.08 | 25.12 |
| 0.4 | Linear | 68.65 | 67.66 | 65.85 | 10.00 | 0.99 | 1.81 | - |
| | Tanh | 73.13 | 73.29 | 69.22 | 32.65 | -0.16 | 4.07 | 36.57 |
| | HardTanh | 73.33 | 72.19 | 69.25 | 29.93 | 1.14 | 2.94 | 39.32 |
| | ReLU | 80.36 | 79.93 | **71.96** | 45.38 | 0.43 | 7.97 | 26.58 |
| | Leaky ReLU | 80.58 | 79.82 | 66.12 | 45.67 | 0.76 | 13.70 | 20.45 |
| | Swish | 79.01 | 77.29 | 70.36 | 47.83 | 1.72 | 6.93 | 22.53 |
| | Mish | 79.57 | 78.59 | 71.30 | **52.16** | 0.98 | 7.29 | 19.14 |
| | GeLU | **81.67** | **80.62** | 72.32 | 48.66 | 1.05 | 8.30 | 23.66 |

Table 11: Server accuracy of `ConvNet4` using four different $\alpha$ and four different participation $R$, where $N = 100$. The most right columns show the accuracy drop as non-IIDness increases. Since using Linear at $\alpha = 0.01, 0.1$ with participation ratio 0.1, 0.2, 0.3, and 0.4 could not train, leave blank at $\alpha = 0.01 \rightarrow 0.1, \alpha = 1 \rightarrow \alpha = 0.1$ with participation ratio 0.1, 0.2, 0.3, and 0.4.

| Participation Ratio | Activation Function | $\alpha = 10$ | $\alpha = 1$ | $\alpha = 0.1$ | $\alpha = 0.01$ | $\alpha = 10 \rightarrow 1$ | $\alpha = 1 \rightarrow 0.1$ | $\alpha = 0.1 \rightarrow 0.01$ |
|---|---|---|---|---|---|---|---|---|
| 0.1 | Linear | 58.24 | 57.38 | 10.00 | 10.00 | 0.86 | - | - |
| | Tanh | 58.98 | 57.86 | **46.61** | 31.97 | 1.12 | 11.25 | 14.64 |
| | HardTanh | **59.85** | **59.62** | 41.90 | 30.34 | 0.23 | 17.72 | 11.56 |
| | ReLU | 51.61 | 49.73 | 41.95 | 28.83 | 1.88 | 7.78 | 13.12 |
| | Leaky ReLU | 51.60 | 49.76 | 42.05 | 28.90 | 1.84 | 7.71 | 13.15 |
| | Swish | 47.01 | 46.28 | 40.00 | 29.88 | 0.73 | 6.28 | 10.12 |
| | Mish | 50.00 | 49.59 | 43.37 | **33.13** | 0.41 | 6.22 | 10.24 |
| | GeLU | 48.64 | 48.02 | 41.50 | 31.61 | 0.62 | 6.52 | 9.89 |
| 0.2 | Linear | 62.48 | 62.43 | 10.00 | 10.00 | 0.05 | - | - |
| | Tanh | 64.49 | 64.14 | 52.58 | 29.50 | 0.35 | 11.56 | 23.08 |
| | HardTanh | **65.27** | **65.40** | 54.43 | 30.09 | -0.13 | 10.97 | 24.34 |
| | ReLU | 57.80 | 56.23 | 48.37 | 34.03 | 1.57 | 7.86 | 14.34 |
| | Leaky ReLU | 57.85 | 56.16 | 48.34 | 33.92 | 1.69 | 7.82 | 14.42 |
| | Swish | 52.62 | 51.48 | 46.16 | 35.65 | 1.14 | 5.32 | 10.51 |
| | Mish | 57.30 | 55.08 | 50.02 | **38.94** | 2.22 | 5.06 | 11.08 |
| | GeLU | 55.59 | 54.34 | 47.46 | 36.09 | 1.25 | 6.91 | 11.37 |
| 0.3 | Linear | 64.66 | 64.46 | 10.00 | 10.00 | 0.20 | - | - |
| | Tanh | 67.34 | 66.57 | **61.79** | 31.35 | 0.77 | 4.78 | 30.44 |
| | HardTanh | **68.35** | **67.86** | 61.75 | 29.93 | 0.49 | 6.11 | 31.82 |
| | ReLU | 62.43 | 60.38 | 50.67 | 37.91 | 2.05 | 9.71 | 12.76 |
| | Leaky ReLU | 62.62 | 60.34 | 50.76 | 38.02 | 2.28 | 9.58 | 12.74 |
| | Swish | 58.04 | 55.88 | 49.45 | 39.78 | 2.16 | 6.43 | 9.67 |
| | Mish | 65.53 | 62.21 | 52.80 | **42.57** | 3.32 | 9.41 | 10.23 |
| | GeLU | 62.39 | 59.20 | 50.61 | 39.14 | 3.19 | 8.59 | 11.47 |
| 0.4 | Linear | 65.71 | 65.83 | 10.00 | 10.00 | -0.12 | - | - |
| | Tanh | 68.83 | 68.37 | **62.61** | 29.49 | 0.46 | 5.76 | 33.12 |
| | HardTanh | **69.95** | **69.35** | 59.45 | 27.22 | 0.60 | 9.90 | 32.23 |
| | ReLU | 66.88 | 64.39 | 53.26 | 40.96 | 2.49 | 11.13 | 12.30 |
| | Leaky ReLU | 67.02 | 64.60 | 53.17 | 41.17 | 2.42 | 11.43 | 12.00 |
| | Swish | 64.78 | 60.13 | 51.72 | 41.86 | 4.65 | 8.41 | 9.86 |
| | Mish | 68.96 | 67.27 | 56.67 | **44.41** | 1.69 | 10.60 | 12.26 |
| | GeLU | 67.59 | 63.42 | 53.35 | 40.99 | 4.17 | 10.07 | 12.36 |

Table 12: Server accuracy of `ConvNet4` using CIFAR-100 as dataset and $N = 100$. The most right columns show the accuracy drop as non-IIDness increases. Since using Linear at $\alpha = 0.01$ with participation ratio 0.1, 0.2, 0.3, and 0.4 could not train, leave blank at $\alpha = 0.01 \to 0.1$ with participation ratio 0.1, 0.2, 0.3, and 0.4.

| Participation Ratio | Activation Function | $\alpha = 10$ | $\alpha = 1$ | $\alpha = 0.1$ | $\alpha = 0.01$ | $\alpha = 10 \to 1$ | $\alpha = 1 \to 0.1$ | $\alpha = 0.1 \to 0.01$ |
|---|---|---|---|---|---|---|---|---|
| 0.1 | Linear | 25.94 | 25.57 | 22.58 | 1.00 | 0.37 | 2.99 | - |
| | Tanh | 26.56 | 26.03 | 21.81 | 13.47 | 0.53 | 4.22 | 8.34 |
| | HardTanh | **29.04** | **28.66** | **23.70** | **14.66** | 0.38 | 4.96 | 9.04 |
| | ReLU | 21.83 | 22.67 | 17.98 | 10.97 | -0.84 | 4.69 | 7.01 |
| | Leaky ReLU | 21.95 | 22.77 | 18.12 | 11.13 | -0.82 | 4.65 | 6.99 |
| | Swish | 19.66 | 20.67 | 15.41 | 11.10 | -1.01 | 5.26 | 4.31 |
| | Mish | 23.75 | 23.30 | 18.42 | 11.86 | 0.45 | 4.88 | 6.56 |
| | GeLU | 21.21 | 21.77 | 16.65 | 11.15 | -0.56 | 5.12 | 5.50 |
| 0.2 | Linear | 31.78 | 31.89 | 28.39 | 1.00 | -0.11 | 3.50 | - |
| | Tanh | 34.01 | 34.74 | 30.75 | 19.56 | -0.73 | 3.99 | 11.19 |
| | HardTanh | **35.68** | **36.08** | **31.76** | **21.93** | -0.40 | 4.32 | 9.83 |
| | ReLU | 26.91 | 27.59 | 23.99 | 15.61 | -0.68 | 3.60 | 8.38 |
| | Leaky ReLU | 26.93 | 27.77 | 24.04 | 15.68 | -0.84 | 3.73 | 8.36 |
| | Swish | 26.39 | 26.14 | 21.55 | 14.57 | 0.25 | 4.59 | 6.98 |
| | Mish | 29.40 | 28.89 | 24.89 | 16.41 | 0.51 | 4.00 | 8.48 |
| | GeLU | 27.66 | 26.73 | 23.26 | 15.08 | 0.93 | 3.47 | 8.18 |
| 0.3 | Linear | 35.50 | 35.32 | 31.47 | 1.00 | 0.18 | 3.85 | - |
| | Tanh | 37.68 | 38.76 | 34.24 | 23.11 | -1.08 | 4.52 | 11.13 |
| | HardTanh | **39.05** | **39.40** | **35.03** | **24.09** | -0.35 | 4.37 | 10.94 |
| | ReLU | 29.81 | 30.24 | 27.31 | 17.92 | -0.43 | 2.93 | 9.39 |
| | Leaky ReLU | 29.74 | 30.26 | 27.45 | 17.97 | -0.52 | 2.81 | 9.48 |
| | Swish | 29.96 | 29.53 | 25.34 | 17.37 | 0.43 | 4.19 | 7.97 |
| | Mish | 32.33 | 32.71 | 29.19 | 20.16 | -0.38 | 3.52 | 9.03 |
| | GeLU | 30.24 | 29.77 | 27.24 | 17.89 | 0.47 | 2.53 | 9.35 |
| 0.4 | Linear | 37.48 | 37.53 | 33.79 | 1.00 | -0.05 | 3.74 | - |
| | Tanh | 40.01 | 40.85 | 37.13 | 26.35 | -0.84 | 3.72 | 10.78 |
| | HardTanh | **41.69** | **41.73** | **37.41** | **27.44** | -0.04 | 4.32 | 9.97 |
| | ReLU | 30.75 | 31.45 | 30.55 | 20.51 | -0.70 | 0.90 | 10.04 |
| | Leaky ReLU | 31.05 | 31.61 | 30.79 | 20.61 | -0.56 | 0.82 | 10.18 |
| | Swish | 32.54 | 32.37 | 27.61 | 19.45 | 0.17 | 4.76 | 8.16 |
| | Mish | 34.64 | 35.94 | 33.11 | 22.01 | -1.30 | 2.83 | 11.10 |
| | GeLU | 32.37 | 32.12 | 29.27 | 20.73 | 0.25 | 2.85 | 8.54 |

Table 13: Server accuracy of `ConvNet4` with four different client participation $R$ (0.4, 0.3, 0.2, 0.1) and fixed Dirichlet constant $\alpha = 0.01$ for additional setting using a learning rate 0.005. We use $N = 20$ and $N = 100$ for the number of clients and CIFAR-10 as the dataset.

| Activation Function | $N = 100$ | | | | $N = 20$ | | | |
|---|---|---|---|---|---|---|---|---|
| | $R = 0.4$ | $R = 0.3$ | $R = 0.2$ | $R = 0.1$ | $R = 0.4$ | $R = 0.3$ | $R = 0.2$ | $R = 0.1$ |
| Linear | 10.00 | 10.00 | 10.00 | 10.00 | 49.63 | 10.00 | 10.00 | 10.00 |
| Tanh | 46.30 | 43.73 | 40.85 | 36.96 | 51.81 | 47.59 | 42.14 | 33.17 |
| HardTanh | **47.92** | **45.06** | **42.31** | **38.30** | **51.86** | **48.69** | **42.44** | **34.06** |
| ReLU | 38.02 | 34.83 | 30.88 | 28.66 | 38.26 | 35.02 | 32.39 | 25.17 |
| Leaky ReLU | 38.17 | 35.15 | 31.13 | 28.74 | 38.53 | 35.19 | 32.59 | 25.32 |
| Swish | 36.15 | 33.72 | 32.05 | 28.43 | 41.93 | 37.57 | 33.68 | 26.32 |
| Mish | 39.72 | 36.94 | 34.67 | 30.56 | 46.11 | 40.88 | 36.88 | 28.32 |
| GeLU | 37.24 | 34.57 | 32.64 | 28.46 | 43.80 | 39.54 | 35.47 | 27.15 |

## C.2 RESNET RESULT

Table 14, Table 15, and Table 16 shows the result of `Resnet20`, `Resnet32`, and `Resnet44` with all combinations of $R$ and $\alpha$ with $N = 100$. For all values of $R$ and $\alpha$, `Resnet20` using HardTanh shows the highest accuracy. According to the existence of shortcut layer's existence, deeper layer can use features that the activation function has excluded via a shortcut (He et al., 2016), which helps to prevent the recent SOTA activation functions' accuracy drop. As a result, using `Resnet32` and `Resnet44`, the recent SOTA activation functions have a small accuracy gap with the Tanh-like activation functions and surpass in some conditions.

Table 14: Server accuracy of `Resnet20` using four different $\alpha$ and four different participation $R$, where $N = 100$. The most right columns show the accuracy drop as non-IIDness increases.

| Participation Ratio | Activation Function | $\alpha = 10$ | $\alpha = 1$ | $\alpha = 0.1$ | $\alpha = 0.01$ | $\alpha = 10 \to 1$ | $\alpha = 1 \to 0.1$ | $\alpha = 0.1 \to 0.01$ |
|---|---|---|---|---|---|---|---|---|
| | Linear | 49.01 | 48.60 | 38.60 | 24.63 | 0.41 | 10.00 | 13.97 |
| | Tanh | 50.68 | 49.20 | 38.83 | 23.82 | 1.48 | 10.37 | 15.01 |
| 0.1 | HardTanh | **51.23** | **49.44** | **39.28** | **23.91** | 1.79 | 10.16 | 15.37 |
| | ReLU | 48.81 | 48.12 | 37.53 | 23.34 | 0.69 | 10.59 | 14.19 |
| | Leaky ReLU | 48.63 | 48.15 | 37.71 | 23.50 | 0.48 | 10.44 | 14.21 |
| | Linear | 56.93 | 56.96 | 45.73 | 26.70 | -0.03 | 11.23 | 19.03 |
| | Tanh | **59.66** | 57.42 | **46.58** | 26.50 | 2.24 | 10.84 | 20.08 |
| 0.2 | HardTanh | 59.60 | **57.54** | 46.19 | **26.52** | 2.06 | 11.35 | 19.67 |
| | ReLU | 56.90 | 56.95 | 45.35 | 25.73 | -0.05 | 11.60 | 19.62 |
| | Leaky ReLU | 56.78 | 56.49 | 45.18 | 26.10 | 0.29 | 11.31 | 19.08 |
| | Linear | 63.17 | 61.67 | 49.35 | 28.25 | 1.50 | 12.32 | 21.07 |
| | Tanh | **64.63** | **61.74** | 49.69 | **27.90** | 2.89 | 12.05 | 21.79 |
| 0.3 | HardTanh | 64.41 | 61.59 | **50.16** | 27.86 | 2.82 | 11.43 | 22.30 |
| | ReLU | 61.99 | 61.59 | 48.71 | 27.58 | 0.40 | 12.88 | 21.13 |
| | Leaky ReLU | 61.87 | 61.34 | 48.38 | 27.38 | 0.53 | 12.96 | 21.00 |
| | Linear | 66.24 | 65.30 | 52.09 | 30.17 | 0.94 | 13.21 | 21.92 |
| | Tanh | **68.08** | **65.05** | 52.92 | 28.92 | 3.03 | 12.13 | 24.00 |
| 0.4 | HardTanh | 67.90 | 64.91 | **53.69** | **29.51** | 2.99 | 11.22 | 24.18 |
| | ReLU | 66.10 | 64.94 | 52.54 | 28.28 | 1.16 | 12.40 | 26.26 |
| | Leaky ReLU | 66.04 | 64.82 | 52.23 | 28.70 | 1.22 | 12.59 | 23.53 |

Table 15: Server accuracy of `Resnet32` using four different $\alpha$ and four different participation $R$, where $N = 100$. The most right columns show the accuracy drop as non-IIDness increases.

| Participation Ratio | Activation Function | $\alpha = 10$ | $\alpha = 1$ | $\alpha = 0.1$ | $\alpha = 0.01$ | $\alpha = 10 \to 1$ | $\alpha = 1 \to 0.1$ | $\alpha = 0.1 \to 0.01$ |
|---|---|---|---|---|---|---|---|---|
| | Linear | 50.06 | 47.94 | 37.39 | 21.95 | 2.12 | 10.55 | 15.44 |
| | Tanh | 49.55 | 47.15 | 38.24 | 21.61 | 2.40 | 8.91 | 16.63 |
| 0.1 | HardTanh | 50.04 | 47.99 | **38.34** | 22.37 | 2.05 | 9.65 | 15.97 |
| | ReLU | 50.68 | 48.39 | 37.87 | **24.29** | 2.29 | 10.52 | 13.58 |
| | Leaky ReLU | **50.71** | **48.57** | 37.33 | 23.91 | 2.14 | 11.24 | 13.42 |
| | Linear | 60.12 | 56.84 | 43.01 | 24.98 | 3.28 | 13.83 | 18.03 |
| | Tanh | **60.03** | 56.81 | 43.92 | 25.90 | 3.22 | 12.89 | 18.02 |
| 0.2 | HardTanh | 59.30 | **57.39** | **43.99** | 25.51 | 1.91 | 13.40 | 18.48 |
| | ReLU | 58.52 | 56.34 | 43.09 | **26.66** | 2.18 | 13.25 | 16.43 |
| | Leaky ReLU | 58.97 | 56.16 | 43.42 | 26.42 | 2.81 | 12.74 | 17.00 |
| | Linear | 64.51 | 62.20 | 46.15 | 28.63 | 2.31 | 16.05 | 17.52 |
| | Tanh | **64.60** | 61.96 | **47.77** | 28.04 | 2.64 | 14.19 | 19.73 |
| 0.3 | HardTanh | 63.74 | **62.20** | 47.73 | 28.03 | 1.54 | 14.47 | 19.70 |
| | ReLU | 64.23 | 61.58 | 46.49 | **28.90** | 2.65 | 15.09 | 17.59 |
| | Leaky ReLU | 64.20 | 61.59 | 46.81 | 28.49 | 2.61 | 14.78 | 18.32 |
| | Linear | 68.45 | 65.75 | 49.49 | 30.28 | 2.70 | 16.26 | 19.21 |
| | Tanh | **67.75** | **66.58** | **51.03** | 29.41 | 1.17 | 15.55 | 21.62 |
| 0.4 | HardTanh | 67.33 | 66.17 | 50.75 | 29.18 | 1.16 | 15.42 | 21.57 |
| | ReLU | 67.52 | 65.55 | 49.72 | 30.18 | 1.97 | 15.83 | 19.54 |
| | Leaky ReLU | 67.46 | 65.71 | 49.40 | **30.43** | 1.75 | 16.31 | 18.97 |

Table 16: Server accuracy of `Resnet44` using four different $\alpha$ and four different participation $R$, where $N = 100$. The most right columns show the accuracy drop as non-IIDness decreases.

| Participation Ratio | Activation Function | $\alpha = 10$ | $\alpha = 1$ | $\alpha = 0.1$ | $\alpha = 0.01$ | $\alpha = 10 \to 1$ | $\alpha = 1 \to 0.1$ | $\alpha = 0.1 \to 0.01$ |
|---|---|---|---|---|---|---|---|---|
| 0.1 | Linear | 49.68 | 47.28 | 39.56 | 23.26 | 2.40 | 7.72 | 16.30 |
| | Tanh | 49.13 | 46.47 | 38.19 | 23.06 | 2.66 | 8.28 | 15.13 |
| | HardTanh | **49.33** | **47.10** | **38.88** | **23.77** | 2.23 | 8.22 | 15.11 |
| | ReLU | 48.89 | 46.38 | 38.57 | 22.84 | 2.51 | 17.81 | 15.73 |
| | Leaky ReLU | 48.77 | 46.27 | 37.35 | 23.05 | 2.50 | 8.92 | 14.30 |
| 0.2 | Linear | 57.55 | 55.98 | 45.14 | 27.50 | 1.57 | 10.84 | 17.64 |
| | Tanh | 57.94 | 55.54 | **46.42** | **27.23** | 2.40 | 9.12 | 19.19 |
| | HardTanh | **58.80** | **57.54** | 45.98 | 26.46 | 1.26 | 11.56 | 19.52 |
| | ReLU | 56.89 | 54.23 | 44.42 | 26.06 | 2.66 | 9.81 | 18.36 |
| | Leaky ReLU | 57.02 | 53.76 | 43.97 | 26.37 | 3.26 | 9.79 | 17.60 |
| 0.3 | Linear | 63.29 | 61.20 | 47.78 | 30.71 | 2.09 | 13.42 | 17.07 |
| | Tanh | **64.43** | 61.27 | 50.67 | **29.53** | 3.16 | 10.60 | 21.14 |
| | HardTanh | 64.31 | **61.42** | **51.23** | 28.84 | 2.89 | 10.19 | 22.39 |
| | ReLU | 62.79 | 60.19 | 46.25 | 28.41 | 2.60 | 13.94 | 17.84 |
| | Leaky ReLU | 62.63 | 59.63 | 46.84 | 28.27 | 3.00 | 12.79 | 18.57 |
| 0.4 | Linear | 67.30 | 65.09 | 51.25 | 32.67 | 2.21 | 13.84 | 18.58 |
| | Tanh | **68.14** | 64.67 | 52.89 | **32.37** | 3.47 | 11.78 | 20.52 |
| | HardTanh | 67.91 | **65.02** | **53.95** | 31.90 | 2.89 | 11.07 | 22.05 |
| | ReLU | 66.74 | 64.20 | 50.31 | 30.51 | 2.54 | 13.89 | 19.80 |
| | Leaky ReLU | 66.41 | 64.26 | 50.40 | 31.03 | 2.15 | 13.86 | 19.37 |

