# OpenReview forum: "Revisiting the Activation Function for Federated Image Classification"
_ICLR.cc/2023/Conference — Submitted to ICLR 2023_

### Official Review · Reviewer_fb3s · 2022-10-24

**Confidence:** 4
**Clarity, Quality, Novelty And Reproducibility:** Clarity and quality is low. Please se…
**Correctness:** 2
**Technical Novelty And Significance:** 1
**Empirical Novelty And Significance:** 1
**Recommendation:** 1

**Strength And Weaknesses:**

Strength:
- The question that the paper sets out to answer empirically is well-defined.
Weaknesses:
-  For an empirical work, the experiments are very limited and have many issues.
- No error bars or multiple runs are used
- The number of datasets that are considered are very limited and it is difficult to be certain if the results generalize.
- The parameters of the training algorithms, e.g. lr, local batch size and steps, local lr, momentum are kept fixed and not tuned for different settings. I suspect this is one main reason that the performance of the models are so low and not on par with what they should be (e.g. for larger resnet models with almost iid, the performance should be >70% in my experience for cifar-10). Due to not well-tuned hyper-parameters for each setting, some weird observations happen, e.g. linear model performs better on cifar-100 compared to cifar-10 in table 1. Another example of this is the observation on 3rd paragraph of page 5.
- Only a limited set of heterogeneity is considered (label heterogeneity) and there are no experiments on different local dataset sizes, covariate shifts, ...
- Section 4 is poorly written and it is full of statements without substantial evidence. Unfortunately, the statements in section 4 are mostly vague, difficult to follow and independent of the results that are presented. In figure 3 and 4, it is not clear what is being presented and how does it relate to the other claims. It seems that learnability and heterogeneity are two different and opposing factors based on Figure 3 and 4. But it is not clear why they should both be in the favor of the tanh like activations.
- Overall, the presented evidence for the claims that the paper makes are not very strong. Moreover, section 4 does not provide a clear picture or intuition on why the claims could be correct either.

**Summary Of The Paper:**

This paper considers the importance of different activation functions in the performance of the centralized and FL trained models. It tries to empirically show that there are considerable differences between the two settings.

**Summary Of The Review:**

The experimental results have many flaws and are not convincing. The claims are not well supported by the observations.

---

> ### Author Response · Authors · 2022-11-18
> **Response**
>
> The tables below shows the result with various hyperparameter settings, the accuracy differs but the order between the Tanh-like activation functions and recent SOTA activation functions does not change. The overall accuracy change with hyperparameter change does not affect the aspect and does not conflict with the claims of this paper. While following some of the optimal hyperparameter settings that other studies have experimented with[1,2,3], we made variations to the learning rate, the number of clients, and the neural network type. Additionally, experimental contexts such as heterogeneity followed previous studies in the same field[4,5].
>
>
> |$\alpha=0.01$|  |	|  | |$\alpha=10$ | | | | |
> |--|--|--|--|--|--|--|--|--|--|
> |lr=0.001|lr_decay|0.2|0.1|0.05|lr=0.001|lr_decay|0.2|0.1|0.005
> ||Tanh|31.81|31.57|36.96||Tanh|40.64|40.15|39.87
> ||HardTanh|**33.11**|**32.72**|**38.37**||HardTanh|**42.80**|**41.70**|**41.27**
> ||ReLU|24.51|23.94|29.17||ReLU|35.81|34.68|34.31
> ||Leaky ReLU|24.53|23.92|29.12||Leaky ReLU|35.83|34.80|34.34
> |lr=0.005|||||lr=0.005||||
> ||Tanh|38.23|31.64|36.87||Tanh|52.96|51.13|50.49
> ||HardTanh|**39.65**|**32.67**|**38.14**||HardTanh|**56.44**|**54.06**|**53.17**
> ||ReLU|30.12|23.89|28.96||ReLU|49.29|47.85|47.02
> ||Leaky ReLU|30.26|23.92|29.02||Leaky ReLU|49.34|47.89|47.06
> |lr=0.01|||||lr=0.01||||
> ||Tanh|**34.54**|**33.41**|23.90||Tanh|61.20|58.79|57.70
> ||HardTanh|32.83|31.25|**24.01**||HardTanh|**61.89**|**60.14**|**59.13**
> ||ReLU|29.48|28.66|10.00||ReLU|53.53|51.65|50.71
> ||Leaky ReLU|29.60|28.90|10.00||Leaky ReLU|53.49|51.70|50.78
> |lr=0.05|||||lr=0.05||||
> ||Tanh|25.97|**33.17**|**22.62**||Tanh|**56.12**|**49.04**|**45.45**
> ||HardTanh|**26.74**|30.56|22.11||HardTanh|52.59|45.60|43.45
> ||ReLU|10.00|29.13|10.00||ReLU|17.23|17.23|17.23
> ||Leaky ReLU|10.00|29.27|10.00||Leaky ReLU|17.57|17.57|17.57
>
>
> |$\alpha=0.01$|||||$\alpha=10$|||||
> |--|--|--|--|--|--|--|--|--|--|
> |lr=0.001|milestone|50,75|50,100|75,150|lr=0.001|milestone|50,75|50,100|75,150|
> ||Tanh|31.57|31.50|32.87||Tanh|40.15|40.46|42.49
> ||HardTanh|**32.72**|**33.02**|**34.34**||HardTanh|**41.70**|**42.25**|**45.03**
> ||ReLU|23.94|24.12|26.51||ReLU|34.68|35.26|39.23
> ||Leaky ReLU|23.92|24.20|26.50||Leaky ReLU|34.80|35.36|39.23
> |lr=0.005|
> ||Tanh|31.64|37.23|39.77||Tanh|51.13|51.75|57.13
> ||HardTanh|**32.67**|**38.70**|**41.13**||HardTanh|**54.06**|**54.88**|**60.65**
> ||ReLU|23.89|29.64|30.49||ReLU|47.85|48.77|51.81
> ||Leaky ReLU|23.92|29.79|30.61||Leaky ReLU|47.89|48.75|51.78
> |lr=0.01|||||lr=0.01|
> ||Tanh|**33.41**|**34.08**|**35.85**||Tanh|58.79|59.65|63.06
> ||HardTanh|31.25|32.13|33.70||HardTanh|**60.14**|**60.80**|**63.59**
> ||ReLU|28.66|28.96|30.41||ReLU|51.65|52.46|55.69
> ||Leaky ReLU|28.90|29.17|30.98||Leaky ReLU|51.70|52.59|55.72
> |lr=0.05|
> ||Tanh|**33.17**|**25.04**|**28.11**||Tanh|**49.04**|**52.31**|**55.97**
> ||HardTanh|30.56|24.26|27.87||HardTanh|45.60|48.79|51.18
> ||ReLU|29.13|10.00|10.00||ReLU|17.23|17.23|17.23
> ||Leaky ReLU|29.27|10.00|10.00||Leaky ReLU|17.57|17.57|17.57
>
> [1] Li, Xiang, et al. "On the convergence of fedavg on non-iid data." arXiv preprint arXiv:1907.02189 (2019).
> [2] Acar, Durmus Alp Emre, et al. "Federated learning based on dynamic regularization." arXiv preprint arXiv:2111.04263 (2021).
> [3] Li, Xiaoxiao, et al. "Fedbn: Federated learning on non-iid features via local batch normalization." arXiv preprint arXiv:2102.07623 (2021).
> [4] Oh, Jaehoon, Sangmook Kim, and Se-Young Yun. "Fedbabu: Towards enhanced representation for federated image classification." arXiv preprint arXiv:2106.06042 (2021).
> [5] Lee, Gihun, et al. "Preservation of the global knowledge by not-true self knowledge distillation in federated learning." arXiv preprint arXiv:2106.03097 (2021).

---

### Official Review · Reviewer_VoE5 · 2022-10-24

**Confidence:** 4
**Correctness:** 2
**Technical Novelty And Significance:** 2
**Empirical Novelty And Significance:** 2
**Recommendation:** 3

**Clarity, Quality, Novelty And Reproducibility:**

The objective of this work is clear, but the experiments are hard to understand and not convincing to support the main contribution.

**Strength And Weaknesses:**

### Strength

- This paper provides an interesting observation that the choice of the activation function plays a critical role in the generalization performance in federate learning, and Tanh-based activation functions outperform Relu-based activation functions, which are widely used in centralized training, in various federated learning scenarios with heterogeneous clients.
- This paper conducts comprehensive experiments that show how each aspect of the federated learning setup (the number of clients, non-IIDness, participation rate, backbone architecture) affects the accuracy according to the different activation functions and provides guidelines for selection activation functions in federated learning.

### Weakness

- The experimental results in Table 3 do not provide convincing evidence demonstrating that Tanh-like activation functions are virtually unaffected by non-IIDness.
    - As illustrated in Table 11, Tanh-like activation functions show a much more significant accuracy drop when the data heterogeneity across clients becomes larger (Dirichlet parameter $\alpha$ gets smaller). This trend becomes more significant when more clients participate per round. While these results support the claim that Tanh-like activation functions are robust to partial participation, but do not support the claim that Tanh-like functions are robust to the non-IIDness of each client.
- The authors claim that the issue of client drift becomes severe since ReLU-based activation functions take fewer features than Tanh-like activation functions. However, Results in Figure 3 and Figure 4 rather show that Tanh-like activation functions have more client drift phenomenon: 1) low CKA similarity between clients, 2) large weight divergence between clients. Note that large weight divergence is known to be a cause of degenerated server accuracy and has been studied in many works [1, 2, 3]. Therefore, these experiments do not support the claim and should provide more convincing evidence.
- Need more additional discussions and experiments:
    - Why are the accuracies of ResNet-based architectures and EfficientNet lower thanConvNet4?
    - 200 rounds of communication seems too short for the convergence of the model. The authors should provide convergence plots or report the results after the model is trained with enough communication rounds.
    - There are no discussions about why large difference in accuracy among ReLU-based activation functions or among Tanh-like activation functions.
    - In Table 4, when $N=20$, why Leaky ReLU performs better with $R=0.3$ than $R=0.4$?
    - ReLU-based activation functions and Tanh-like activation functions may have different optimal learning rates. Authors should compare models trained with each optimal local learning rate.


### Reference
[1] S.P. Karimireddy et al., SCAFFOLD: Stochastic Controlled Averaging for Federated Learning, ICML, 2020.

[2] D.A.E. Acar et al., Federated Learning Based on Dynamic Regularization, ICLR, 2021.

[3] L. Gao et al., FedDC: Federated Learning with Non-IID Data via Local Drift Decoupling and Correction, CVPR, 2022.

**Summary Of The Paper:**

This paper analyzes the effect of various activation functions on server accuracy in federated learning with heterogeneous clients. It shows that Tanh-based activation functions outperform ReLU-based activation functions in most cases, providing the guidelines for selecting activation functions in various federated learning scenarios. This paper also conducts a few experiments to investigate why Tanh-based activation functions are more robust to client heterogeneity than other activation functions.

**Summary Of The Review:**

This paper lacks contribution because (1) the main experiments are difficult to understand, missing many discussions, and do not support few claims, (2) analysis of model behavior does not support the main claim that the Tanh-based activation functions are robust to data heterogeneity of clients, and (3) the paper does not propose any method to handle the issue of existing activation functions in federated learning.

---

> ### Author Response · Authors · 2022-11-18
> **Response**
>
> Answer to Q1 : The accuracy drop of α=0.01 shows due to training failure. For this we conducted an additional experiment using learning rate 0.005 which is shown in Table 12.
>
> Answer to Q2 : The client drift can’t be removed using each client’s heterogeneous data. But we claim that Tanh-like activation functions minimize the effect of client drift during aggregation step. As a result, Figure 3 and Figure 4 shows the difference of feature and weight using the server model which is already converged. Which intend that during the training procedure, important features are excluded using ReLU-based activation functions and loses the learning ability.
>
> Answer to Q3:
> - Q : Why are the accuracies of ResNet-based architectures and EfficientNet lower thanConvNet4?
> - A : ConvNet4 has larger number of parameters compared to ResNet-based architectures and EfficientNet.
>     - ConvNet4 : 3,537,060
>     - ResNet20 : 269,722
>     - Resnet32 : 464,154
>     - Resnet44 : 658,586
>     - MobileNetv2 : 2,236,682
> #
> - Q : 200 rounds of communication seems too short for the convergence of the model. The authors should provide convergence plots or report the results after the model is trained with enough communication rounds.
> - A : The setting using 200 rounds is widely used in FL settings. For making sure I attach the url for the learning curve with N=20, R=0.4, $\alpha=0.1$, Leaky ReLU as the activation function, and ConvNet4 as the model. [URL for the Accuracy Curve.](https://anonymous.4open.science/r/FL_ACT-160B/fig/accuracy_curve.png)
> #
> - Q : There are no discussions about why large difference in accuracy among ReLU-based activation functions or among Tanh-like activation functions.
> - A : See section 4.1. During the FL aggregation step, a problem arises where important features for the global optimum cannot be selected due to client drift. This phenomenon appears to be severe when the recent SOTA activation functions are used. Due to the shape of their activation functions, the excluded features are greater in number than for the Tanh-like activation functions, and a severe accuracy drop occurs.
> #
> - Q : In Table 4, when N=20, why Leaky ReLU performs better with R=0.3 than R=0.4?
> - A : The value has been written wrong. The actual accuracy is 72.27.
> #
> - Q : ReLU-based activation functions and Tanh-like activation functions may have different optimal learning rates. Authors should compare models trained with each optimal local learning rate.
> - A : Table below shows the result using various learning rate decay and learning rate with R=0.1, α=0.01 and N=100.
>
> |$\alpha=0.01$|  |	|  | |$\alpha=10$ | | | | |
> |--|--|--|--|--|--|--|--|--|--|
> |lr=0.001|lr_decay|0.2|0.1|0.05|lr=0.001|lr_decay|0.2|0.1|0.005
> ||Tanh|31.81|31.57|36.96||Tanh|40.64|40.15|39.87
> ||HardTanh|**33.11**|**32.72**|**38.37**||HardTanh|**42.80**|**41.70**|**41.27**
> ||ReLU|24.51|23.94|29.17||ReLU|35.81|34.68|34.31
> ||Leaky ReLU|24.53|23.92|29.12||Leaky ReLU|35.83|34.80|34.34
> |lr=0.005|||||lr=0.005||||
> ||Tanh|38.23|31.64|36.87||Tanh|52.96|51.13|50.49
> ||HardTanh|**39.65**|**32.67**|**38.14**||HardTanh|**56.44**|**54.06**|**53.17**
> ||ReLU|30.12|23.89|28.96||ReLU|49.29|47.85|47.02
> ||Leaky ReLU|30.26|23.92|29.02||Leaky ReLU|49.34|47.89|47.06
> |lr=0.01|||||lr=0.01||||
> ||Tanh|**34.54**|**33.41**|23.90||Tanh|61.20|58.79|57.70
> ||HardTanh|32.83|31.25|**24.01**||HardTanh|**61.89**|**60.14**|**59.13**
> ||ReLU|29.48|28.66|10.00||ReLU|53.53|51.65|50.71
> ||Leaky ReLU|29.60|28.90|10.00||Leaky ReLU|53.49|51.70|50.78
> |lr=0.05|||||lr=0.05||||
> ||Tanh|25.97|**33.17**|**22.62**||Tanh|**56.12**|**49.04**|**45.45**
> ||HardTanh|**26.74**|30.56|22.11||HardTanh|52.59|45.60|43.45
> ||ReLU|10.00|29.13|10.00||ReLU|17.23|17.23|17.23
> ||Leaky ReLU|10.00|29.27|10.00||Leaky ReLU|17.57|17.57|17.57

---

### Official Review · Reviewer_2hFV · 2022-10-25

**Confidence:** 3
**Correctness:** 3
**Technical Novelty And Significance:** 3
**Empirical Novelty And Significance:** 3
**Recommendation:** 6

**Clarity, Quality, Novelty And Reproducibility:**

Clarity and Quality:

Overall very good. Some confusion in the analysis section.

Novelty:

Novel question to explore as far as I am aware of. Though it might be argued that this paper doesn't propose new algorithms for FL.

**Strength And Weaknesses:**

Strength:

1: The overall idea is important and novel. I am not aware of other papers studying the effect of activation function in FL settings. And I am pretty surprised that activation functions can cause significant performance change in FL setting.

2: The experiments seems to be solid and extensive. Authors consider several settings and different algorithms.

Weakness:

1: I am quite confused by the analysis, especially section 4.1. In the first half of 4.1, I quote "This can be summarized simply by saying that the Tanh-like activation functions have low sensitivity to the accuracy drop in the FL aggregation step because they exclude a much smaller number of features than do the recent SOTA activation functions." Up to this point of the section, it seems to be only an intuition instead of explanation. Next, authors measure the CKA similarity and weight similarity between clients. They seem to say ReLU shows higher similarity in both cases compared with Tanh. My question is: if Tanh excludes fewer features in each client, shouldn't it show higher similarity in both CKA (the output features) and weight? From another perspective, as far as I understand, the client drift means each local client learns a significantly different model. Thus, if Tanh somehow alleviates client drift as author claims, shouldn't it result in similar models in all the clients?

2: I think if there is some theoretical analysis on the activation function and how it affects the convergence, it will really strengthen the paper.


**Summary Of The Paper:**

In Federated Learning, there is little attention on how different types activation function affect the final accuracy. In this paper, authors discover that good activation function in centralized training is not good in FL settings and vice versa. Authors further perform some analysis on the weight parameters, latent representations and loss landscape resulted from different activation functions. They show that good activation functions in FL are the ones that exclude fewer features and has smoother landscape.

**Summary Of The Review:**

I recommend weak accept of this paper. I think the overall question is important and novel. Though more work should be focused on analysis section and maybe some theoratical analysis.

---

> ### Author Response · Authors · 2022-11-18
> **Response**
>
> Answer to the question “If Tanh excludes fewer feature in each client, shouldn’t it show higher similarity in both CKA (the output features) and weight?”.
> - Similarity of CKA and weight differs due to training with heterogenous dataset on each client. Because Tanh excludes fewer feature it changes more than ReLU with different data and CKA and weight differs more.
>
> Answer to the question “The client drift means each local client learns a significantly different model. Thus, if Tanh somehow alleviates client drift as author claims, shouldn’t it result in similar models in all the clients”.
> - Using Tanh, the server model is trained to alleviate client drift, however due to heterogeneous client data, finetuning by each client data differs the model to client-friendly. The model using Tanh does not mean it can perfectly train a global model which fits to every client.
>
> For addition, theoretical analysis is very difficult even in conventional deep learning. Inasmuch as convergence of the ReLU in deep network is still difficult to see, there is no clear theoretical basis for this or other latest activation functions.

---

### Official Review · Reviewer_H9Ky · 2022-10-30

**Confidence:** 3
**Clarity, Quality, Novelty And Reproducibility:** The paper is well-written and easy to…
**Correctness:** 3
**Technical Novelty And Significance:** 2
**Empirical Novelty And Significance:** 3
**Recommendation:** 5

**Strength And Weaknesses:**

Overall, the paper is well-written and easy to follow. The experimental results demonstrate that the proposed method is promising. The authors provide a guideline for selecting activation functions in FL, an explanation for the performance degradation and a novel activation function HardTanh.

However, my concern is whether the reason why HardTanh works so well has something to do with poor training. I see that the authors use a learning decay of 0.1 at rounds 50 and 75, which might be beneficial for some activation functions. I suggest that the authors employ multiple
training strategies to observe the result. And more additional experiments lead us to believe that the benefits of the HardTanh activation functions are independent of model training. Moreover, it will be more convincing to compare it with the latest SOTA models.

**Summary Of The Paper:**

This study clarifies that the drop in accuracy varies according to the activation function in the FL. Due to the shape of the function, the accuracy of the SOTA activation function drops in the FL setting and HardTanh outperforms other activation functions in most environments.
Furthermore, the authors provide guidelines and benchmark data for choosing activation functions in various FL settings.

**Summary Of The Review:**

See Strength And Weaknesses

---

> ### Author Response · Authors · 2022-11-18
> **Response**
>
> |$\alpha=0.01$|  |	|  | |$\alpha=10$ | | | | |
> |--|--|--|--|--|--|--|--|--|--|
> |lr=0.001|lr_decay|0.2|0.1|0.05|lr=0.001|lr_decay|0.2|0.1|0.005
> ||Tanh|31.81|31.57|36.96||Tanh|40.64|40.15|39.87
> ||HardTanh|**33.11**|**32.72**|**38.37**||HardTanh|**42.80**|**41.70**|**41.27**
> ||ReLU|24.51|23.94|29.17||ReLU|35.81|34.68|34.31
> ||Leaky ReLU|24.53|23.92|29.12||Leaky ReLU|35.83|34.80|34.34
> |lr=0.005|||||lr=0.005||||
> ||Tanh|38.23|31.64|36.87||Tanh|52.96|51.13|50.49
> ||HardTanh|**39.65**|**32.67**|**38.14**||HardTanh|**56.44**|**54.06**|**53.17**
> ||ReLU|30.12|23.89|28.96||ReLU|49.29|47.85|47.02
> ||Leaky ReLU|30.26|23.92|29.02||Leaky ReLU|49.34|47.89|47.06
> |lr=0.01|||||lr=0.01||||
> ||Tanh|**34.54**|**33.41**|23.90||Tanh|61.20|58.79|57.70
> ||HardTanh|32.83|31.25|**24.01**||HardTanh|**61.89**|**60.14**|**59.13**
> ||ReLU|29.48|28.66|10.00||ReLU|53.53|51.65|50.71
> ||Leaky ReLU|29.60|28.90|10.00||Leaky ReLU|53.49|51.70|50.78
> |lr=0.05|||||lr=0.05||||
> ||Tanh|25.97|**33.17**|**22.62**||Tanh|**56.12**|**49.04**|**45.45**
> ||HardTanh|**26.74**|30.56|22.11||HardTanh|52.59|45.60|43.45
> ||ReLU|10.00|29.13|10.00||ReLU|17.23|17.23|17.23
> ||Leaky ReLU|10.00|29.27|10.00||Leaky ReLU|17.57|17.57|17.57
>
> The table above shows the result using various learning rate decay and learning rate with R=0.1, α=0.01 and N=100.
> The table below shows the result using various milestones and learning rate decay with R=0.1, α=0.01 and N=100.
> With various hyperparameter settings, the accuracy differs but the order between the Tanh-like activation functions and recent SOTA activation functions does not change. The overall accuracy change with hyperparameter change does not affect the aspect and does not conflict with the claims of this paper. For more experiments refer to the Appendix.
>
> |$\alpha=0.01$|||||$\alpha=10$|||||
> |--|--|--|--|--|--|--|--|--|--|
> |lr=0.001|milestone|50,75|50,100|75,150|lr=0.001|milestone|50,75|50,100|75,150|
> ||Tanh|31.57|31.50|32.87||Tanh|40.15|40.46|42.49
> ||HardTanh|**32.72**|**33.02**|**34.34**||HardTanh|**41.70**|**42.25**|**45.03**
> ||ReLU|23.94|24.12|26.51||ReLU|34.68|35.26|39.23
> ||Leaky ReLU|23.92|24.20|26.50||Leaky ReLU|34.80|35.36|39.23
> |lr=0.005|
> ||Tanh|31.64|37.23|39.77||Tanh|51.13|51.75|57.13
> ||HardTanh|**32.67**|**38.70**|**41.13**||HardTanh|**54.06**|**54.88**|**60.65**
> ||ReLU|23.89|29.64|30.49||ReLU|47.85|48.77|51.81
> ||Leaky ReLU|23.92|29.79|30.61||Leaky ReLU|47.89|48.75|51.78
> |lr=0.01|||||lr=0.01|
> ||Tanh|**33.41**|**34.08**|**35.85**||Tanh|58.79|59.65|63.06
> ||HardTanh|31.25|32.13|33.70||HardTanh|**60.14**|**60.80**|**63.59**
> ||ReLU|28.66|28.96|30.41||ReLU|51.65|52.46|55.69
> ||Leaky ReLU|28.90|29.17|30.98||Leaky ReLU|51.70|52.59|55.72
> |lr=0.05|
> ||Tanh|**33.17**|**25.04**|**28.11**||Tanh|**49.04**|**52.31**|**55.97**
> ||HardTanh|30.56|24.26|27.87||HardTanh|45.60|48.79|51.18
> ||ReLU|29.13|10.00|10.00||ReLU|17.23|17.23|17.23
> ||Leaky ReLU|29.27|10.00|10.00||Leaky ReLU|17.57|17.57|17.57

---

### Author Response · Authors · 2022-11-18
**General Response**

Thank you for reviewing our paper of your personal time and attention. We have addressed your review and comment on additional experiments and claims.

We respond to the reviewer's issues in the following. We summarize the main changes as fllows:
- We performed an additional experiment for different settings of hyperparameter.
- Additional explanation of analysis.

---

### Decision · Program_Chairs · 2023-01-20

**Decision:**

Reject

**Justification For Why Not Higher Score:**

Paper has a lot of flaws and does not merit a higher score

**Justification For Why Not Lower Score:**

N/A

**Metareview: Summary, Strengths And Weaknesses:**

In this work, the effectiveness of activation functions in various federated learning settings is verified. It is observed that off-the-shelf activation functions that are used in centralized settings exhibit a different performance trend than in federated settings. HardTanh is found to achieve the best accuracy when severe data heterogeneity or low participation rate is present. The authors also provide an analysis to investigate why the representation powers of activation functions are changed in a federated setting by measuring the similarities in terms of weight parameters and representations. Guidelines for selecting activation functions in cross-silo and cross-device settings are also provided. The reviewers thought the paper was easy to follow and thought the numerical experiments are promising. The reviewers however raised a variety of concerns including that: (1) "whether the reason why HardTanh works so well has something to do with poor training", (2) confusion about the analysis of Sec 4.1 (3) limited experiments (4) no error bars (5) limited number of datasets (6) issues with parameter choices. The authors provided a response but this response does not address all of the concerns raised by the reviewers in a satisfactory manner. It is clear the paper requires a substantial review and therefore I recommend rejection.